# Research on furniture image classification based on MobileNetNAK

Danyang Zhang[1], Yi Zhai[2,3], Peiyuan Li[1], Fan Yang[4] and Runpeng Du[1]

[1] Daegu University, Daegu, Republic of South Korea
[2] Shandong Academy of Sciences, Qilu University of Technology, Jinan, Shandong, China
[3] Shandong Provincial Key Laboratory of Computing Power Internet and Service Computing, Jinan, Shandong, China
[4] Linyi Vocational University of Science and Technology, Linyi, Shandong, China

## ABSTRACT

With the rapid development of the furniture industry, automatic classification of furniture images has become an important research area. However, this task faces several challenges, including complex image backgrounds, diverse furniture types, and varying forms. To address these issues, we propose a novel furniture image classification method, MobileNetNAK, based on the MobileNetV3 network. First, the method integrates a non-local attention module to capture non-local dependencies within images, significantly enhancing the model's ability to extract key information. Second, the Adamax optimizer is employed to train the model. By adaptively adjusting the learning rate, it accelerates convergence and reduces the risk of overfitting. Third, the Kolmogorov–Arnold networks method is incorporated to decompose complex convolution operations into multiple simpler ones, thereby improving computational efficiency and feature extraction capabilities. Experimental results demonstrate that MobileNetNAK significantly improves classification performance in furniture image tasks. On Dataset 1, the model achieves improvements of 6.7%, 6.6%, 6.6%, and 6.6% in accuracy, precision, recall, and F1-score, respectively, compared to the baseline. On Dataset 2, the corresponding improvements are 2.7%, 2.4%, 2.7%, and 2.9%. Additionally, the model maintains a high inference speed of 147.80 fps, balancing performance with computational efficiency. These results highlight the strong adaptability and deployment potential of MobileNetNAK in multi-category and fine-grained furniture image classification tasks, offering a novel and effective solution for this domain.

## INTRODUCTION

Image classification, a fundamental and essential task in computer vision, seeks to precisely recognize and categorize the elements within an image using algorithmic models (*Binder, Müller & Kawanabe, 2012*). As technology continues to evolve, this technique has found extensive applications across numerous domains, significantly contributing to the advancement of intelligent systems (*Chen et al., 2021*). In the home furnishing industry, furniture classification, as an important part of promoting intelligent transformation, has become increasingly prominent in the context of growing personalized and customized needs. Visual data of furniture, as a rich and insightful medium, not only conveys essential

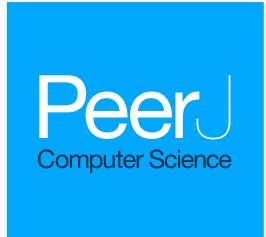

Corresponding author
Peiyuan Li, q81874138@outlook.com

details about the design, materials, and aesthetics but also encapsulates valuable insights into market trends, consumer tastes, and innovative design. Precise extraction and strategic application of this information are crucial for enhancing the design workflow, boosting manufacturing efficiency, and precisely aligning with market needs. Therefore, exploring efficient and accurate furniture image classification technology has become the key to promoting the intelligent upgrading of the home furnishing industry and achieving high-quality development (*Ye et al., 2022*).

The critical role of furniture image categorization in the smart home sector is growing increasingly vital, driven by the rising consumer desire for customized and high-quality living spaces. Although furniture image data, as an important source of visual data, provides a rich information base for accurate classification and personalized recommendation of furniture, this process also faces many complexity and technical problems. The variety of furniture, with its wide array of shapes, intertwines design elements, materials, and styles, resulting in highly diverse and complex imagery (*Hu et al., 2017*). Secondly, the quality and clarity of furniture images are frequently compromised by various elements, including the perspective from which they are captured, the lighting during photography, and disruptions in the background. These aspects significantly complicate the process of categorizing such images. Moreover, for the classification of furniture imagery, it is crucial to balance both precision and speed. This means that while maintaining high standards of accuracy, the system must also be capable of swiftly delivering results to meet the demands of online home goods retailers and physical stores (*Manavis et al., 2024*). Consequently, addressing these technical hurdles and achieving a streamlined, precise classification of furniture images is pivotal for advancing the industry's intelligence and improving customer satisfaction.

In the field of home design and manufacturing, furniture image classification is a basic and crucial task. It not only helps designers and manufacturers to better understand and organize products, but also provides consumers with a more accurate and personalized shopping experience. With the continuous development of technology, furniture image classification methods have also evolved from traditional methods to deep learning methods.

Conventional approaches to classifying furniture images predominantly hinge on manually crafted features and classification algorithms. This process typically encompasses two primary phases: the extraction of features and the categorization itself. During the feature extraction phase, experts apply their specialized knowledge and techniques to distill essential attributes like color, texture, and form from the images. These extracted characteristics then serve as the foundation for the classifiers, which are tasked with identifying the type of furniture. Among the frequently utilized classifiers are k-nearest neighbors (k-NN) (*Abeywickrama, Cheema & Taniar, 2016*), support vector machines (SVM) (*Cortes, 1995*), and random forests (RF) (*Mitchell & Mitchell, 1997*). In literature (*Chao & Li, 2022*), K-NN distance entropy was used to screen remote sensing images. In literature (*Adugna, Xu & Fan, 2022*), the performance of RF and SVM algorithms in large

area land cover mapping in some areas of Africa was tested by using coarse resolution images, and good classification results were obtained. Although these methods have achieved certain results in some scenarios, they rely heavily on the characteristics of artificial design, which not only requires rich professional knowledge, but also may not fully capture the complexity and diversity of furniture images.

Over the past few years, the swift advancement in deep learning, particularly with the emergence of sophisticated models like convolutional neural networks (CNNs) (*Krizhevsky, Sutskever & Hinton, 2012*), has led to significant progress in the classification of furniture images. Reference *Kilic et al. (2023)* proposed a novel 2D CNN architecture with fewer convolutional layers to achieve high accuracy and excellent performance in wood tree species classification. Reference *Dong et al. (2022)* used CNN to achieve a good classification of hyperspectral images. The deep learning method can automatically learn and extract advanced features from the original data without manual intervention. This approach enhances both the precision and speed of categorization while also lessening the reliance on specialized expertise. In the field of furniture image classification, deep learning models such as AlexNet (*Krizhevsky, Sutskever & Hinton, 2012*), Visual Geometry Group (VGG) (*Simonyan & Zisserman, 2014*), Residual Network (ResNet) (*He et al., 2016*) have been widely used and achieved significant performance improvement. These models can gradually extract the deep features of furniture images by stacking convolution layer, pooling layer and fully connected layer, so as to achieve more accurate classification. Compared with traditional algorithms, deep learning algorithms have significantly improved the accuracy and efficiency of classification in furniture image classification tasks by automatically learning advanced features, effective computation, robust adaptability, and the potential to significantly enhance performance when integrated with various technologies, leading to groundbreaking advancements in furniture image categorization.

Despite the progress achieved in furniture image classification, several critical challenges remain insufficiently addressed. First, most existing approaches primarily focus on coarse-grained category or style recognition, while lacking effective modeling of fine-grained features such as wear patterns, aging traces, and structural details. Second, although lightweight network architectures offer advantages in computational efficiency, they often involve a trade-off in classification accuracy, particularly when dealing with multi-class, style-diverse furniture datasets. Furthermore, there is a noticeable lack of systematic exploration and integration of attention mechanisms, optimization strategies, and nonlinear feature modeling techniques within current methods. Therefore, this study aims to bridge the gap in improving fine-grained classification accuracy of lightweight models by addressing key limitations in feature attention, training stability, and nonlinear representation learning.

To address the challenges in furniture image classification—namely insufficient feature extraction, limited model stability, and inadequate representation of complex visual patterns—this article proposes a lightweight deep learning model, MobileNetNAK, based on the MobileNetV3 architecture and integrated with Normalization-based Attention

Module (NAM), Adamax, and Kolmogorov-Arnold Networks (KANs) modules. While maintaining model efficiency, MobileNetNAK significantly improves classification performance. The main contributions of this work are as follows:

(1) Integration of a Normalization-based Attention Module (NAM) to enhance the model's sensitivity to key regions in furniture images, such as wear traces, local aging textures, and structural details. NAM combines both channel and spatial attention mechanisms with batch normalization scaling factors, guiding the model to focus on discriminative regions and effectively mitigating the representational limitations of traditional lightweight models in handling fine-grained features.

(2) Adoption of the Adamax optimizer to improve training stability and convergence efficiency. In light of the challenges posed by imbalanced class distributions and high-dimensional feature noise in complex furniture scenes, Adamax leverages the infinity norm to estimate the second moment of gradients, thereby stabilizing learning rates and enhancing the model's robustness and generalization capability under complex conditions.

(3) Incorporation of the Kolmogorov-Arnold Network (KAN) to strengthen the model's ability to capture complex nonlinear features. Unlike traditional linear convolutions, KANs employ learnable one-dimensional spline mappings to replace fixed activation functions, effectively enhancing the model's expressiveness in capturing high-level semantic variations such as style, material, and structural changes in furniture images, thus further improving classification accuracy.

By integrating these three key techniques, MobileNetNAK achieves superior performance in extracting discriminative information from furniture images and demonstrates state-of-the-art accuracy on two challenging benchmark datasets, offering a high-precision and cost-effective solution for intelligent visual recognition in the furniture industry.

The remainder of this article is structured in the following manner. 'Related Work' delves into the relevant literature, while 'Framework' provides an in-depth overview of the MobileNetNAK architecture. 'Experiments' presents the specifics of the experiments conducted and their outcomes. The article concludes with a summary of the findings in 'Conclusions'.

## RELATED WORK

### Overview of traditional furniture image classification methods

The initial phase of conventional approaches in categorizing furniture images involves the extraction of key attributes. This process is designed to distill distinctive and informative elements from the images, enabling the following classification steps to be more precise. Typically, this includes analyzing color, texture, and form as the primary means to capture these essential characteristics (*Mingqiang, Kidiyo & Joseph, 2008*).

Color attributes primarily rely on statistical elements, such as the color histogram and color moment of furniture images, for classification. These elements can illustrate the

distribution and variation of colors within furniture images, which is crucial for differentiating furniture with distinct color styles (*Chen, Liu & Chen, 2010*). For texture attributes, methods like the Gray-Level Co-occurrence Matrix (GLCM) (*Dhingra & Bansal, 2020*) and Local Binary Pattern (LBP) (*Haralick, Shanmugam & Dinstein, 1973*) are employed to extract the surface texture information of furniture. These attributes can capture the fine details and texture variations on the furniture surface, playing a key role in distinguishing furniture that has similar colors but different textures (*Ojala, Pietikainen & Maenpaa, 2002*). Shape attributes, on the other hand, are derived through techniques such as edge detection and contour extraction, providing measurements like perimeter, area, and compactness. These features can reflect the overall shape and contour information of furniture, and play an important role in distinguishing furniture with different shapes and structures (*Kas, Ruichek & Messoussi, 2021*).

Following the extraction of features, conventional approaches necessitate the creation of suitable classifiers for categorization. SVM, rooted in statistical learning theory, transforms furniture feature vectors into a high-dimensional space to facilitate classification through the use of support vectors and kernel functions. While SVM boasts high accuracy and robust generalization, it can encounter significant computational complexity with large datasets. Decision trees and random forests leverage decision tree structures to build classification models, enhancing performance through ensemble learning. Decision trees are straightforward and quick, but they can be prone to overfitting. Random forests, by aggregating multiple decision trees and using a voting mechanism, enhance both the stability and precision of classifications. K-NN classifies based on the proximity of furniture feature vectors in the feature space, offering simplicity and ease of understanding without the need for training. However, K-NN can be computationally intensive with large datasets and may not perform well with high-dimensional data.

Although traditional methods have achieved certain results in furniture image classification tasks, feature extraction is highly dependent on the experience and expertise of researchers, which may lead to incomplete or biased feature selection. This will affect the classification performance and accuracy of subsequent classifiers. Secondly, the traditional classifier has limited generalization ability in the face of complex and changeable furniture images. Traditional classifiers are often designed and optimized based on specific assumptions and models, which are difficult to adapt to large-scale data sets and complex and changeable furniture images.

## Overview of deep learning furniture image classification methods

The technology for categorizing furniture images has long been a critical area of study within the home furnishings sector. Conventional methods have been utilized for many years, offering a basic approach to organizing and searching for furniture. However, these methods often fall short in terms of precision and sophistication. The advent of advanced artificial intelligence, particularly the surge in deep learning, has opened up fresh possibilities for this domain. Deep learning algorithms can autonomously derive intricate features from extensive datasets of furniture images, enabling highly accurate and intelligent classification.

On the one hand, classical CNN architectures such as AlexNet, VGG, ResNet are widely used, and their multi-layer convolution and pooling operations can automatically extract image features for accurate classification. Reference *Wan et al. (2024)* introduced a one-dimensional convolutional neural network that integrates near-infrared spectroscopy with deep learning methods to identify different types of wood. In *Hu et al. (2017)*, an improved furniture image automatic classification model DGOVGG16 was proposed by combining deep grouping over-parametric convolution and VGG16 model, with an average accuracy of 95.51%.

In the field of furniture image classification, the techniques for feature extraction and fusion play a crucial role. The use of multi-modal feature extraction has gained significant attention. Besides conventional visual attributes, multi-modal feature vectors are now being created by integrating physical properties such as material composition and dimensions. By employing deep learning models for fusion, a more thorough representation of furniture characteristics can be achieved, thereby enhancing classification accuracy. In one study (*Yang et al., 2022*), micro-CT technology was utilized to capture detailed images of cross, radial, and tangential sections of 24 Pterocarpus species. This approach, combined with an extreme learning machine, enabled highly effective and precise classification of these wood types. Another research (*Li et al., 2022*) introduced a multi-level feature extraction framework designed to boost the descriptive power of point cloud data, resulting in superior classification and semantic segmentation on various standard datasets.

Data augmentation techniques play a crucial role in addressing the limited size of furniture image datasets. Conventional approaches to data augmentation encompass a variety of transformations, such as random cropping, rotation, flipping, and scaling. These modifications help to diversify the dataset and boost the generalization capabilities of deep learning models. For instance, by randomly rotating and flipping furniture images, the model can learn to recognize features from multiple angles, thereby enhancing its classification accuracy across different viewpoints. Moreover, Generative Adversarial Networks (GANs) (*Goodfellow et al., 2020*) have become increasingly popular for data augmentation. A study (*Suh et al., 2021*) introduced a classification-enhanced GAN designed to improve performance in scenarios with imbalanced data. This framework consists of three distinct networks: a generator, a discriminator, and a classifier, which collectively outperform standard data augmentation methods in unbalanced conditions. Another research (*Yu & Liu, 2021*) presented a novel GAN specifically for augmenting and enhancing wafer images, tackling issues related to class imbalance and insufficient labeled data. This GAN can produce synthetic images that closely resemble real furniture, thus expanding the dataset. By integrating these synthetic images with actual ones during training, the model's performance is significantly enhanced, enabling it to better classify various types of furniture, including different styles and materials.

Model optimization and compression technology have promoted the development of deep learning models in furniture image classification. Numerous algorithms have been developed to enhance the speed and precision of model training processes. Variables such as stochastic gradient descent (SGD) (*Duchi, Hazan & Singer, 2011*), adaptive gradient

algorithm (Adagrad), Adadelta, root mean square propagation (RMSProp) and adaptive moment estimation (Adam) (*Kingma, 2014*) can adaptively adjust the learning rate to speed up the convergence. Algorithms that leverage second-order derivatives, like the Newton and quasi-Newton methods, can refine parameters with greater precision, thereby enhancing overall performance. In *Xue, Tong & Neri (2022)*, an integrated algorithm of differential evolution and Adam was proposed to improve the global and local search ability by parallel evolution of two subpopulations. Experiments show that the algorithm not only has strong search ability, but also has fast convergence speed.

*Shi (2023)* proposed an improved bilinear convolutional neural network model that incorporates a spatial attention mechanism and dual pooling operations to achieve fine-grained recognition of furniture image styles. The model achieved a recognition accuracy of 76.4% on the FashionStyle14 dataset, representing a 2-percentage-point improvement over the original model while significantly reducing the number of parameters and computational complexity. *Tian, Zhao & Li (2025)* developed an efficient automatic furniture image classification model based on an enhanced VGG16 architecture, integrated with depthwise group over-parameterized convolution (DGOPC). By introducing Rectified Linear Unit (ReLU) and Leaky-ReLU activation functions, the model optimized the training process and improved classification performance.

In the field of visual attention, several advanced attention-based networks have recently emerged to further enhance feature representation capabilities. Cascaded Visual Attention Network (CVANet) (*Zhang et al., 2024*) employs a cascade of multiple attention modules, significantly improving pixel-level detail reconstruction and achieving notable performance gains in image super-resolution tasks. In addition, Generate Adversarial-driven Cross-Aware Network (GACNet) (*Zhang et al., 2023*) integrates a semi-supervised generative adversarial framework with cross-perception attention modules, enabling joint attention to spectral, spatial, and textural features, and has demonstrated superior performance in recognition tasks.

Deep learning has obvious advantages in furniture image classification tasks. First, it can automatically learn the high-level feature representation of furniture images, overcome the subjectivity and incompleteness of traditional feature extraction methods, and have stronger adaptability and robustness in the face of complex and changeable furniture images. Secondly, the deep learning model exhibits robust generalization capabilities. Once trained on extensive datasets, it can effectively capture the overarching feature representations and classification principles of various furniture types. This allows it to precisely categorize and recognize new furniture images that it has not encountered before.

## FRAMEWORK

As a lightweight convolutional neural network model launched by Google, MobileNetV3 is designed for mobile devices and embedded systems (*Howard et al., 2019*). The overall structure of MobileNetV3 is shown in Fig. 1. It inherits the core idea of the MobileNet series—deep separable convolution. This convolution technique substantially decreases the computational load and the quantity of model parameters, enabling the model to sustain high performance while drastically cutting down on computational demands and

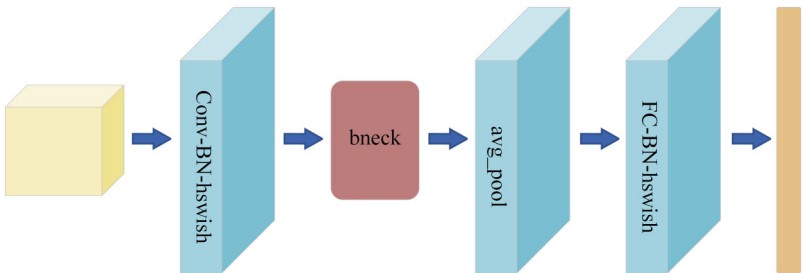

**Figure 1 MobileNetV3 overall structure diagram.**

memory usage. Furthermore, MobileNetV3 has incorporated innovative design approaches, including the integration of the Squeeze-and-Excitation module from Squeeze-and-Excitation Network (SENet) and the introduction of updated activation functions like h-swish. These enhancements allow for dynamic adjustment of the importance of each channel in the feature map, boosting the network's representational power. This, in turn, facilitates more efficient information propagation during the training phase, leading to faster convergence and improved generalization.

Because of its high efficiency, light weight and high precision, MobileNetV3 has become an ideal choice for furniture image classification tasks, which meets the requirements of real-time, accuracy and portability. The application of MobileNetV3 in furniture image classification is expected to promote technological progress and application development, and support the intelligence and automation of the furniture industry.

Based on the MobileNetV3 algorithm, we innovatively propose a new furniture image classification algorithm MobileNetNAK by skillfully combining advanced technical means such as NAM, Adamax and KANs. The algorithm not only inherits the advantages of MobileNetV3, such as high efficiency, lightweight and high precision, but also further enhances the capture ability and classification accuracy of the model for furniture image features. Figure 2 shows the overall framework of MobileNetNAK algorithm, which clearly shows the collaborative work between modules and jointly promotes the performance improvement of furniture image classification tasks.

## NAM

The Normalization-based Attention Module (NAM) (*Liu et al., 2021*) is an innovative attention mechanism design. By introducing normalization operations, it aims to allocate attention weights more effectively, thereby improving the model's ability to extract and utilize key features of input data (such as images, texts, *etc.*). The primary goal is to address the challenges that arise during feature selection and data integration in conventional attention mechanisms. NAM integrates channel attention with spatial attention to achieve this, and uses the scaling factor of batch normalization (BN) (*Ioffe, 2015*) to measure the importance of channels and pixels, so as to achieve effective recognition and utilization of features.

The key aspect of NAM lies in its distinctive attention computation, which leverages normalization and integrates both channel and spatial attention. Unlike conventional

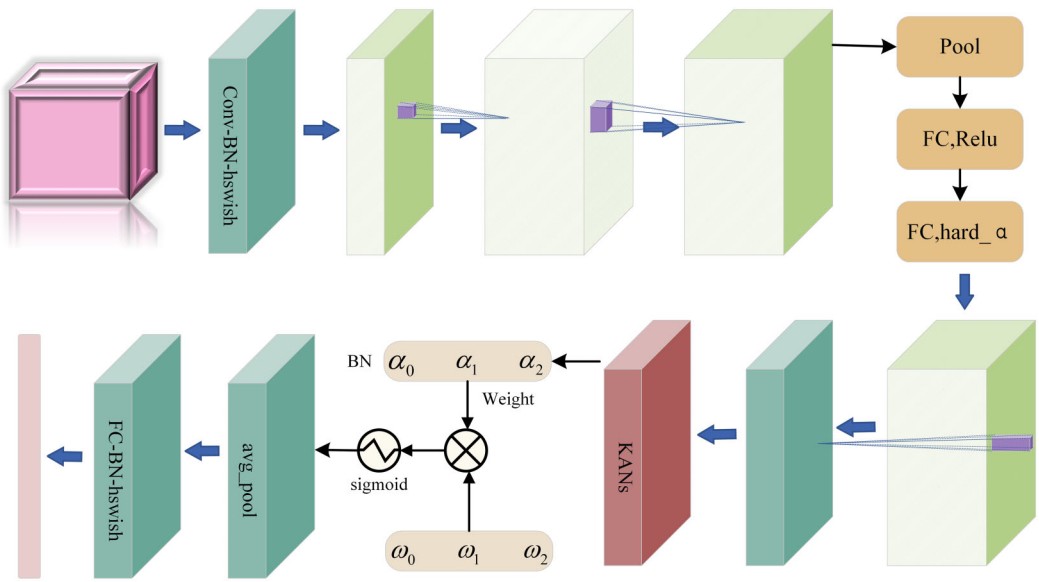

**Figure 2  MobileNetNAK overall structure diagram.**

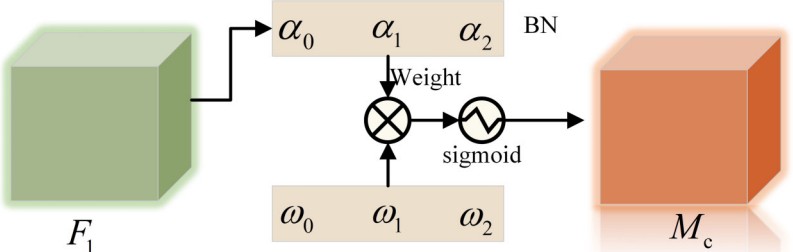

**Figure 3  Channel attention sub-module.**

attention methods, NAM refrains from employing fully connected and convolutional layers, thereby enhancing computational efficiency. Furthermore, NAM enforces a sparsity constraint on the attention weights, diminishing less important weights through a regularization term in the loss function. This approach bolsters the model's generalization and efficiency. By merging two sub-modules that address channel and pixel-level features, NAM can more effectively capture a wide range of feature information, leading to improved accuracy in feature extraction and classification.

The channel attention sub-module is shown in Fig. 3, $\alpha$ is the scale factor of each channel, the weight $w_i$ is shown in Formula (1), and the output feature $M_c$ is shown in Formula (2).

$$w_i = \frac{\alpha_i}{\sum_{j=0} \alpha_j} \tag{1}$$

$$M_c = sigmoid(W_\alpha(BN(F_1))). \tag{2}$$

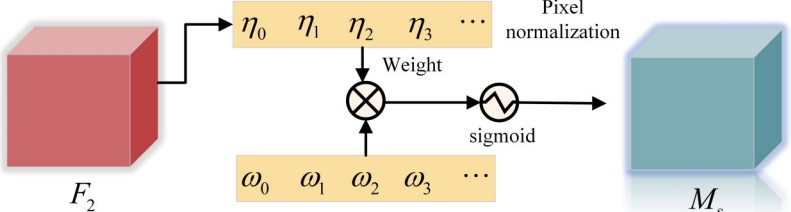

**Figure 4 Spatial attention sub-module.**

The spatial attention sub-module is shown in Fig. 4, λ is the scale factor, the weight $w_i$ is shown in Formula (3), and the output feature $M_s$ is shown in Formula (4).

$$w_i = \frac{\eta_i}{\sum_{j=0} \eta_j} \tag{3}$$

$$M_s = sigmoid\left(W_\eta(BN(F_2))\right). \tag{4}$$

With its high efficiency, accuracy and unique attention mechanism, NAM has become the preferred technology in furniture image classification tasks. Not only does it substantially boost the precision of classification and strengthen the model's resilience, but it also diminishes the computational expenses and simplifies integration and scalability. Therefore, we choose to use NAM to bring more efficient and accurate solutions to furniture image classification tasks through its advantages.

In furniture image classification, NAM enables the model to more accurately capture key features such as texture, color, and shape. This method improves the model's ability to identify essential characteristics, thereby enhancing classification accuracy. It substantially elevates the accuracy of classification, strengthens the model's resilience, and simultaneously decreases computational expenses. Additionally, it is straightforward to integrate and scale. Therefore, we choose to use NAM to bring more efficient and accurate solutions to furniture image classification tasks through its advantages.

### Adamax optimizer

The Adam algorithm is equivalent to introducing the temporary gradient idea of the Nestrov momentum method into the adaptive moment estimation algorithm. In each calculation of the gradient, a temporary update of the parameter is obtained first. After the parameter is temporarily updated, the temporary gradient is calculated. The initial momentum and the subsequent momentum are gauged using a provisional gradient. The provisional values of the initial and subsequent momentums are then utilized to determine the parameter adjustments.

The Adamax optimizer is an advanced version of the Adam optimization algorithm. Adam integrates the benefits of the Nesterov momentum technique and the RMSProp method, adapting the learning rate for each parameter through the computation of the first and second moment estimates of the gradient. Adamax builds upon this by specifically refining the handling of the learning rate's upper limit.

The essence of the Adamax optimizer lies in altering the learning rate adjustment mechanism within the Adam algorithm. The detailed Adam algorithm can be found in *Kingma (2014)*. Adamax uses an infinite norm to estimate the second moment (*i.e.*, variance) of the gradient, thus obtaining a more conservative (*i.e.*, not too fast) learning rate update strategy.

$V_t$ and $q_t$ are the first moment (mean) and the second moment (non-central variance) estimates of the gradient, respectively. We use $h_t$ to denote the infinite norm constrained $q_t$:

$$h_t = \zeta_2^{\infty} q_{t-1} + \left(1 - \zeta_2^{\infty}\right)|s_t|^{\infty}. \tag{5}$$

Substitute it into the Adam update Eq. (2) and replace it with $h_t$ to get the Adamax update rule as shown in Type 3:

$$\vartheta_{t+1} = \vartheta_t - \frac{\eta}{\sqrt{\hat{q}_t + \varepsilon}}\hat{V}_t \tag{6}$$

$$\vartheta_{t+1} = \vartheta_t - \frac{\eta}{h_t}\hat{V}_t. \tag{7}$$

The characteristics and advantages of the Adamax optimizer are that it uses the infinite norm for second-order moment estimation, thereby achieving effective control of the upper bound of the learning rate and avoiding the problem of training stagnation caused by too fast reduction of the learning rate. In contrast to Adam and other optimization methods, Adamax demonstrates greater reliability during training, particularly when handling extensive datasets and intricate models. In addition, the sensitivity of Adamax to hyperparameters (such as learning rate, decay rate, *etc.*) is relatively low, which enables it to maintain strong applicability in different tasks and models, and provides an efficient and reliable optimization method for deep learning model training.

We utilize the Adamax optimizer for the furniture image classification, which substantially enhances the stability and efficiency of the training. It effectively addresses issues such as slow convergence and the tendency to get stuck in local optima during the training process. By dynamically adjusting the learning rate for each parameter, Adamax can speed up model convergence and mitigate the risk of settling into suboptimal solutions. In addition, the low sensitivity of Adamax to hyperparameters also makes it more robust in furniture image classification tasks, and can maintain stable performance under different data sets and model architectures. Therefore, the use of Adamax optimizer can further improve the accuracy and generalization ability of furniture image classification tasks.

## Kolmogorov-arnold networks

KANs is inspired by the Kolmogorov-Arnold theorem, which states that if f is a multivariate continuous function on a bounded domain, then f can be written as a finite combination of univariate continuous function addition binary operations. This representation theorem provides a solid theoretical basis for the structural design of neural networks, indicating that complex multivariable functions can be represented by a combination of univariate functions.

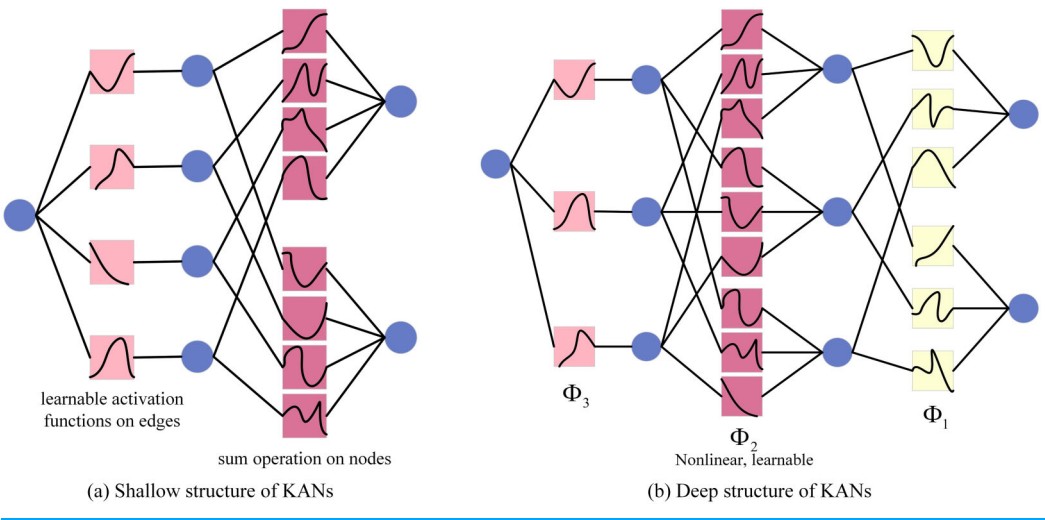

(a) Shallow structure of KANs                    (b) Deep structure of KANs

**Figure 5  KANs structure diagram.**               

Similar to Multi-Layer Perceptron (MLP), KANs also have a fully connected structure. However, the main difference between them is the application location of the activation function. In MLP, activation functions are placed on nodes (neurons), while in KANs, activation functions are placed on edges (weights), and these activation functions are learnable. Therefore, the KANs network has no linear weight matrix at all, and each weight parameter is replaced by a learnable one-dimensional spline function. Specifically, the nodes of KANs only sum the input signals without any nonlinear processing. Each weight parameter (*i.e.*, the edge) is represented by a learnable one-dimensional spline function that maps the input directly to the output without the need for intermediate weighted sum and subsequent universal activation. The KANs structure diagram is shown in Fig. 5.

In KANs, the initial input vector is fed into the network, after which every input variable is distributed to each neuron within the hidden layer. Each of these neurons in the hidden layer computes the individual function output for all the input variables it receives and aggregates these outputs. A univariate activation function is then utilized to introduce nonlinearity. Subsequently, the outputs from all the hidden layer neurons are combined through a weighted sum to produce the final output. The resultant output for the shallow KANs model is detailed in Eq. (8), while the deep KANs model's output is provided in Eq. (8).

$$h(x) = h(x_1, \ldots, x_n) = \sum_{m=1}^{2n+1} \phi \left( \sum_{v=1}^{n} \phi_{m,v}(x_v) \right). \tag{8}$$

The above formula is for smooth $h : [0,1]^n \to \mathbb{R}$, where $\varnothing_{m,v} : [0,1]$ and

$$KAN(x) = (\phi_{I-1} {}^{\circ} \phi_{I-2} {}^{\circ} \ldots {}^{\circ} \phi_1 {}^{\circ} \phi_0) x \tag{9}$$

where $\Phi_i$ is the function matrix corresponding to the layer i KAN, the general KAN network is a combination of layer I, given an input vector $x_0 \in \mathbb{R}^{n_0}$.

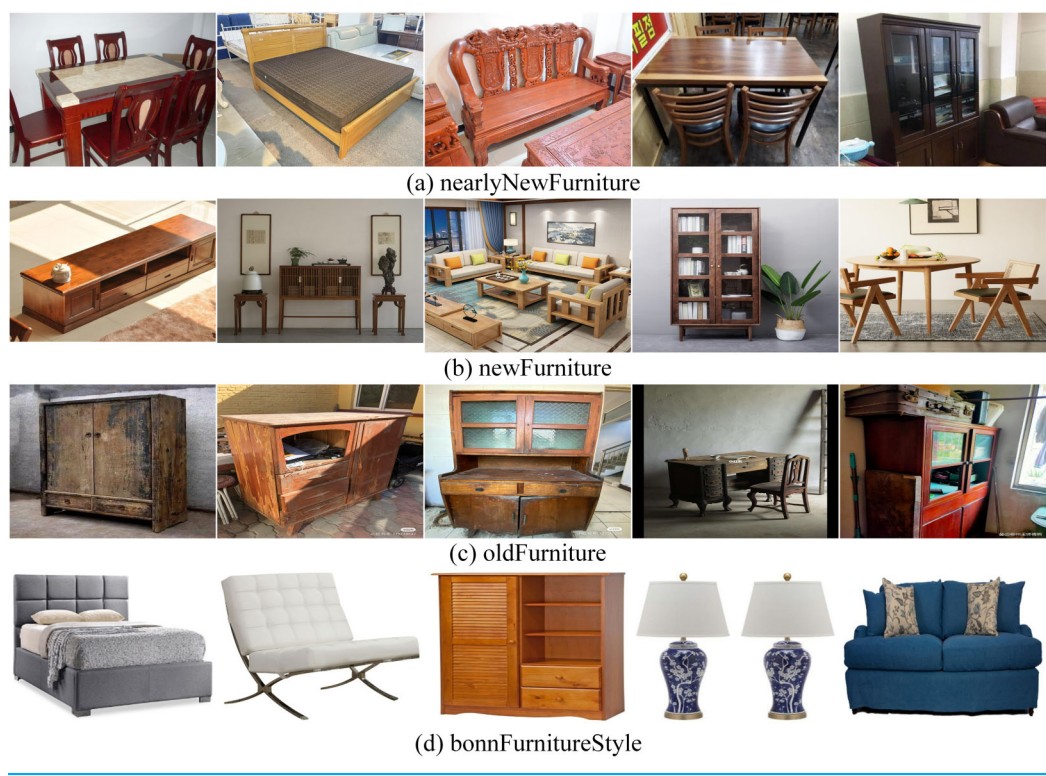

(a) nearlyNewFurniture

(b) newFurniture

(c) oldFurniture

(d) bonnFurnitureStyle

**Figure 6 Data set example diagram.**

KANs are introduced to further enhance the feature extraction ability of MobileNetV3, especially when dealing with complex image data. KANs can solve the problem that traditional convolution operations may have insufficient feature extraction capabilities when dealing with complex image data. By introducing new convolution operations or network structures, KANs can learn richer feature representations, thereby improving the classification performance of the model. At the same time, KANs may also help the model maintain high recognition performance when dealing with furniture images with complex texture, shape or color changes.

# EXPERIMENTS

## Datasets and evaluation metrics

The furniture image classification task in this study involves two datasets. Dataset 1 includes furniture of varying ages, categorized into three classes as shown in Fig. 6: (A) nearlyNewFurniture, with 317 images; (B) newFurniture, with 325 images; and (C) oldFurniture, with 303 images. The new furniture images were primarily sourced from brand-new furniture stores and online retailers, with the selected images featuring furniture without any signs of use. The nearly new furniture images were collected from the second-hand market, focusing on items with a freshness rating of over 90%, showing slight signs of use but no visible damage. The old furniture images were obtained from waste disposal stations and second-hand furniture markets, showcasing items with significant signs of use and visible damage. Dataset 2 is the Bonn Furniture Style dataset

(*Aggarwal et al., 2018*), as shown in Fig. 6D, which includes six types of furniture: bed, chair, vanity, lamp, sofa, and table, and encompasses 17 different furniture styles. For the experimental dataset, approximately 1,000 images were randomly selected from each of the 17 styles, resulting in a total of 6,592 images.

The two datasets were divided into training, testing, and validation sets in an 8:1:1 ratio, with the validation set used during training and the testing set for model evaluation. The datasets exhibit a relatively balanced distribution of samples across categories, and the images cover a wide variety of furniture styles and designs. This ensures stability and fairness during the training and evaluation process, while also providing a rich and reliable data foundation for model validation experiments.

The proposed MobileNetNAK algorithm was comprehensively evaluated through comparisons with a variety of baseline methods to validate its performance in the task of furniture image classification. The baseline models include Locally Adaptive Network-based Support Vector Machine (LA-NSVM), Vision Restoration Transformer (VRT), Spatial Uncertainty Network (SUNet), Semantic Knowledge Distillation (SKD), Residual Attention Network (RAN), ResNet18, Lzy University of North Carolina at Charlotte (LzyUNCC), M-Net, MobileNetV3, and Self-Interaction Network (SINet), covering traditional methods, lightweight networks, and mainstream deep learning architectures.

In addition, to further assess the generalization and effectiveness of the proposed improvements, we integrated the three core components of the NAK module into three widely adopted backbone architectures—Xception, ResNet18, and EfficientNet—resulting in the XceptionNAK, ResNet18NAK, and EfficientNetNAK models, respectively. A comparative analysis was then conducted between these models and MobileNetNAK. The performance changes before and after the integration of the NAK module across different architectures further demonstrate the adaptability and efficacy of the proposed MobileNetNAK method in furniture image classification tasks.

In this experiment, we use a variety of evaluation indicators to comprehensively evaluate the performance of each algorithm for the furniture image classification task. These indicators include accuracy, precision, recall and F1-score. These indicators are used to measure the consistency between the predicted results of the classification model and the actual labels, as well as the comprehensive performance of the model in the furniture image classification task.

Accuracy is an important index to measure the consistency between the prediction results of the classification model and the actual labels, which aims to reflect the proportion of the correctly predicted samples in the total samples. Its calculation formula is:

$$Accuracy = \frac{TP + TN}{TP + FP + TN + FN}. \tag{10}$$

Among them, TP represents the sample that is predicted to be true and is actually true, FP represents the sample that is predicted to be true and is actually false, TN represents the sample that is predicted to be false and is actually false, FN represents the sample that is predicted to be false and is actually true. In the task of furniture image classification, the accuracy rate reflects the overall classification ability of the model for all categories of

furniture images, and is the basic index to evaluate the performance of the model. For example, if the model has a high accuracy in classifying furniture such as chairs, tables and sofas, it shows that the model can better identify these furniture categories.

The precision rate is a measure of the proportion of the model that is actually positive in the samples that are predicted to be positive. It reflects the Precision of the model to predict the positive class, and avoids misjudging the negative class samples as the positive class. Its calculation formula is:

$$Precision = \frac{TP}{TP + FP}. \tag{11}$$

In the classification of furniture images, the precision rate is particularly suitable for scenes that focus on reducing false positives. For example, when classifying chairs, the high precision rate means that the model rarely misjudges the table or sofa as a chair. This is very important to ensure the reliability of the classification results, especially in application scenarios that require high precision, such as furniture recommendation systems.

Recall rate is a measure of the proportion of models that are correctly predicted to be positive in samples that are actually positive. It reflects the model's ability to identify positive samples and avoid false negatives. Its calculation formula is:

$$Recall = \frac{TP}{TP + FN}. \tag{12}$$

In furniture image classification, the recall rate is especially suitable for scenes that focus on reducing underreporting. For example, when classifying tables, the high recall rate means that the model can identify most of the samples that are actually tables and avoid missing important classification results. This is very important to ensure the integrity of the classification results, especially in application scenarios that require full coverage, such as furniture inventory management systems.

F1-score is the harmonic mean of precision and recall, which is used to comprehensively evaluate the performance of the model. It strikes a balance between precision and recall to avoid the one-sidedness of a single indicator. Its calculation formula is:

$$F1\text{-}score = 2 \times \frac{Precision \times Recall}{Precision + Recall}. \tag{13}$$

In the task of furniture image classification, F1-score is suitable for scenes that need to focus on both precision and recall. For example, when classifying sofas, a high F1-score means that the model performs well in reducing both false positives and false negatives, balancing the accuracy and completeness of the classification. This is very important for application scenarios that require comprehensive performance, such as the user search experience in furniture classification applications. In addition, to evaluate the computational efficiency of the model, we introduce frames per second (FPS) as an evaluation metric.

Our experiment is completed in the Windows11 system. The algorithm is implemented in Python language on the PyCharm 2022 platform.

## Experimental environment and settings

The experiments were conducted on a Windows 11 operating system using the PyCharm 2022 environment and implemented in Python 3.9. The hardware platform consisted of a computer equipped with an Intel(R) Core(TM) i7-7500U CPU @ 2.70 GHz and 8 GB of RAM, without GPU acceleration. The deep learning framework used was PyTorch 1.13.1, with key dependencies including NumPy 1.24.2, torchvision 0.14.1, matplotlib 3.7.0, seaborn 0.12.2, and tqdm 4.65.0. To ensure reproducibility, a fixed random seed of 42 was used, and Xavier uniform initialization was applied to all model parameters.

The model architecture was based on MobileNetV3-Large. During the experiments, we loaded MobileNetV3 pretrained weights from the ImageNet dataset and applied a transfer learning strategy. Specifically, the original feature extraction structure was retained, while the classification head was restructured, and the entire network was fine-tuned end-to-end. The parameters of the feature extraction part were not entirely frozen, allowing them to participate in backpropagation to improve adaptability to the target task.

All input images in the training, validation, and testing phases were preprocessed consistently using resizing (256 × 256) and center cropping (224 × 224), without applying any random data augmentation strategies. During training, confusion matrices were used to compute per-class accuracy, precision, recall, and F1-score, providing a comprehensive evaluation of model performance. The final evaluation was performed using the model that achieved the highest validation accuracy, ensuring the reliability and stability of the experimental results.

## Comparative experiments

Figures 7 and 8 present the comparative experimental results on Dataset 1 and Dataset 2, respectively, using point-line plots to visualize the performance of different algorithms across four evaluation metrics: accuracy, precision, recall, and F1-score. The horizontal axis represents the compared algorithms, while the vertical axis denotes the metric values, ranging from 0.60 to 0.95.

As shown in Fig. 7, with the continuous evolution of model architectures, all four metrics demonstrate a consistent upward trend. Traditional methods such as LA-NSVM and VRT show relatively low performance, with F1-scores of 0.643 and 0.648, respectively. In contrast, deep learning-based models like ResNet18, MobileNetV3, and SINet exhibit substantial improvements. Upon integrating the proposed NAK modules, models such as ResNet18NAK, XceptionNAK, EfficientNetNAK, and especially MobileNetNAK achieve further enhancements. Among them, MobileNetNAK records the highest performance across all four metrics (Accuracy = 0.904, F1-score = 0.901), indicating the strong compatibility between the proposed improvements and the lightweight backbone architecture.

Figure 8 illustrates the results on Dataset 2, which involves more categories and greater stylistic variation, thereby posing a more challenging classification task. It can be observed that traditional methods still fall significantly behind deep learning models, while lightweight or efficient architectures such as MobileNetV3 and SINet achieve relatively strong performance. With the integration of the NAK modules, most models exhibit
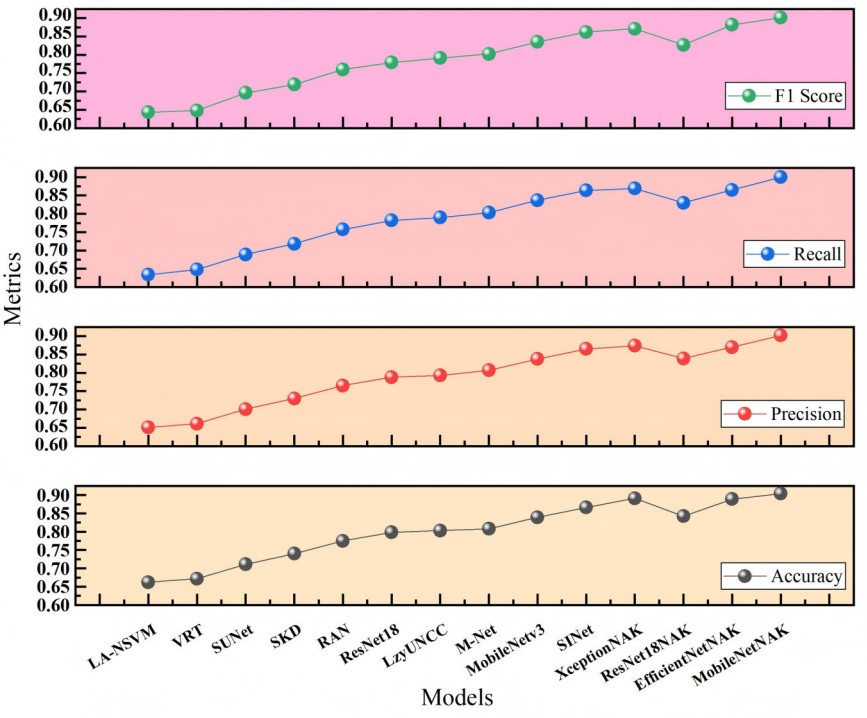

**Figure 7  Performance comparison of different methods on Dataset 1.**

noticeable performance gains. In particular, MobileNetNAK achieves the best overall performance with accuracy and F1-score reaching 0.948, markedly surpassing other models. These results highlight the synergistic effectiveness of the NAM attention mechanism, Adamax optimizer, and KANs architecture in handling complex visual tasks, significantly enhancing the model's ability to perceive fine-grained multi-class features and improving classification precision.

Tables 1 and 2 present the comparative experimental results of multiple models on the Furniture Age Classification Dataset (Dataset 1) and the Bonn Furniture Style Dataset (Dataset 2), respectively.

On Dataset 1, traditional models such as LA-NSVM and VRT show limited performance, with F1-scores of only 0.643 and 0.648, respectively. In contrast, deep convolutional models like ResNet18 and M-Net achieve substantial improvements, reaching F1-scores of 0.779 and 0.802. As a lightweight backbone network, MobileNetV3 further improves performance with an F1-score of 0.835. Upon integrating the proposed NAK modules, several mainstream networks such as XceptionNAK and EfficientNetNAK exhibit varying degrees of performance enhancement. Among them, MobileNetNAK achieves the best results with an accuracy of 0.904, outperforming all other compared models. This confirms the high compatibility between the proposed method and the MobileNetV3 architecture.

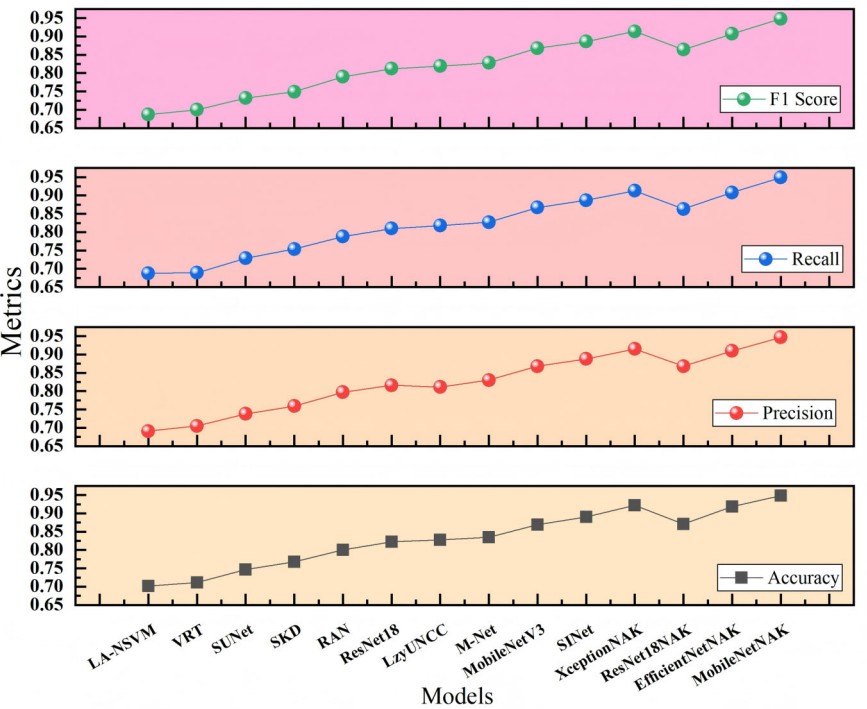

**Figure 8 Performance comparison of different methods on Dataset 2.**

**Table 1 Method comparison on Dataset 1 (Furniture age classification dataset).**

| Models/Metrics | Accuracy | Precision | Recall | F1-score |
|---|---|---|---|---|
| LA-NSVM | 0.662 | 0.651 | 0.634 | 0.643 |
| VRT | 0.672 | 0.661 | 0.648 | 0.648 |
| SUNet | 0.711 | 0.701 | 0.689 | 0.696 |
| SKD | 0.740 | 0.730 | 0.718 | 0.719 |
| RAN | 0.775 | 0.765 | 0.757 | 0.760 |
| ResNet18 | 0.798 | 0.788 | 0.782 | 0.779 |
| LzyUNCC | 0.803 | 0.793 | 0.790 | 0.791 |
| M-Net | 0.808 | 0.807 | 0.803 | 0.802 |
| MobileNetV3 | 0.839 | 0.838 | 0.837 | 0.835 |
| SINet | 0.866 | 0.865 | 0.864 | 0.862 |
| XceptionNAK | 0.891 | 0.874 | 0.869 | 0.871 |
| ResNet18NAK | 0.842 | 0.839 | 0.830 | 0.827 |
| EfficientNetNAK | 0.889 | 0.870 | 0.865 | 0.882 |
| MobileNetNAK | 0.904 | 0.902 | 0.900 | 0.901 |

**Table 2 Method comparison on Dataset 2 (Bonn furniture style dataset).**

| Models/Metrics | Accuracy | Precision | Recall | F1-score |
|---|---|---|---|---|
| LA-NSVM | 0.702 | 0.691 | 0.688 | 0.687 |
| VRT | 0.711 | 0.705 | 0.689 | 0.700 |
| SUNet | 0.747 | 0.738 | 0.729 | 0.732 |
| SKD | 0.768 | 0.759 | 0.754 | 0.749 |
| RAN | 0.800 | 0.797 | 0.788 | 0.790 |
| ResNet18 | 0.822 | 0.816 | 0.810 | 0.812 |
| LzyUNCC | 0.828 | 0.811 | 0.818 | 0.819 |
| M-Net | 0.835 | 0.830 | 0.827 | 0.828 |
| MobileNetV3 | 0.869 | 0.868 | 0.867 | 0.868 |
| SINet | 0.890 | 0.888 | 0.887 | 0.886 |
| XceptionNAK | 0.922 | 0.915 | 0.913 | 0.914 |
| ResNet18NAK | 0.871 | 0.868 | 0.863 | 0.864 |
| EfficientNetNAK | 0.918 | 0.910 | 0.908 | 0.907 |
| MobileNetNAK | 0.948 | 0.947 | 0.949 | 0.948 |

On the more challenging Dataset 2, the baseline MobileNetV3 already demonstrates strong performance with an accuracy of 0.869 and an F1-score of 0.868. After incorporating the NAK modules, XceptionNAK and EfficientNetNAK achieve further improvements, reaching F1-scores of 0.914 and 0.907, respectively. Notably, the proposed MobileNetNAK model again achieves the highest performance with an accuracy and F1-score of 0.948, demonstrating robust and stable performance in the multi-class furniture style recognition task.

Taken together, the results from both datasets indicate that the proposed modules offer complementary advantages: enhancing the model's ability to focus on key visual regions, stabilizing the training process, and improving nonlinear feature representation. These enhancements significantly boost the model's effectiveness in fine-grained furniture image classification and demonstrate strong adaptability and generalization across different task settings.

## Ablation experiments

Figure 9 illustrates the impact of different module combinations on classification accuracy on the Furniture Age Classification Dataset (Dataset 1). The x-axis represents four model configurations: MobileNetV3, MobileNetV3-NAM, MobileNetV3-NAM-Adamax, and the final model MobileNetNAK. As shown, model performance improves progressively with the integration of each proposed component. The baseline MobileNetV3 model achieves an accuracy of 0.837. With the addition of the NAM module, accuracy significantly increases to 0.874, indicating that NAM effectively enhances the model's ability to focus on critical features such as wear and aging patterns. Incorporating the

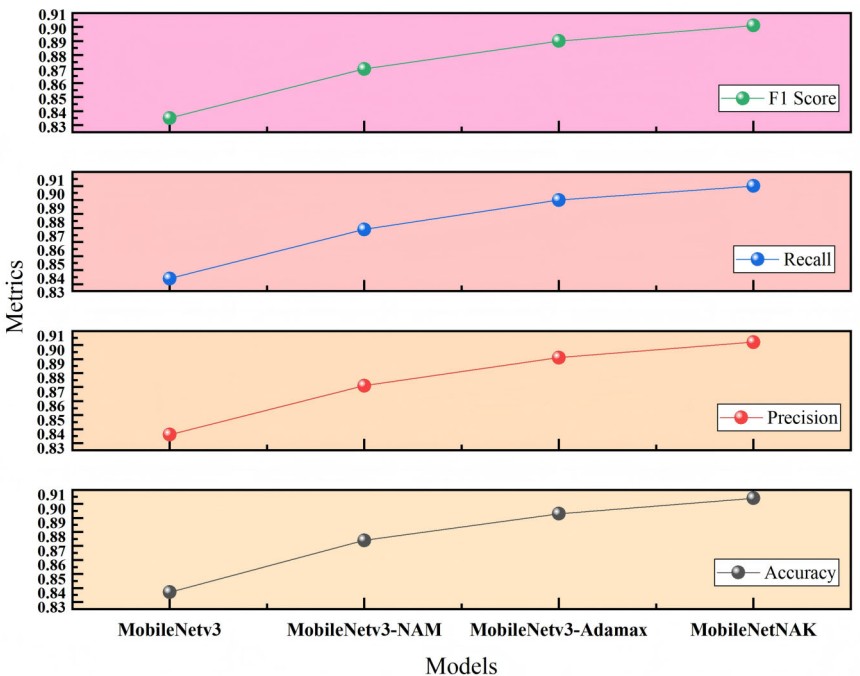

**Figure 9 Ablation study results on Dataset 1.**

Adamax optimizer further improves the accuracy to 0.893, demonstrating its effectiveness in stabilizing the training process and accelerating convergence. Finally, the integration of the KANs module leads to the highest accuracy of 0.904, validating the synergistic effect of the three modules in modeling complex visual patterns.

To ensure the reliability of the results, each configuration was evaluated over three independent runs, and the average performance metrics were reported. Furthermore, paired t-tests were conducted to assess statistical significance. The results show that the performance improvements of MobileNetNAK over the baseline MobileNetV3 in terms of both accuracy and F1-score are statistically significant ($p < 0.01$), indicating that the enhancements are not due to random variation.

Figure 10 presents the ablation study on the Bonn Furniture Style Dataset (Dataset 2), which involves a more complex classification task with a greater number of categories and more pronounced stylistic differences. On this dataset, the baseline MobileNetV3 achieves an accuracy of 0.921. After incorporating all three modules, the proposed MobileNetNAK model reaches the highest accuracy of 0.948. Results obtained from repeated experiments confirm that the three proposed techniques—NAM, Adamax, and KANs—exhibit strong generalization capability and complementary advantages in recognizing fine-grained features across diverse furniture styles. Their integration significantly enhances the model's classification accuracy and stability, particularly under complex and visually challenging scenarios.

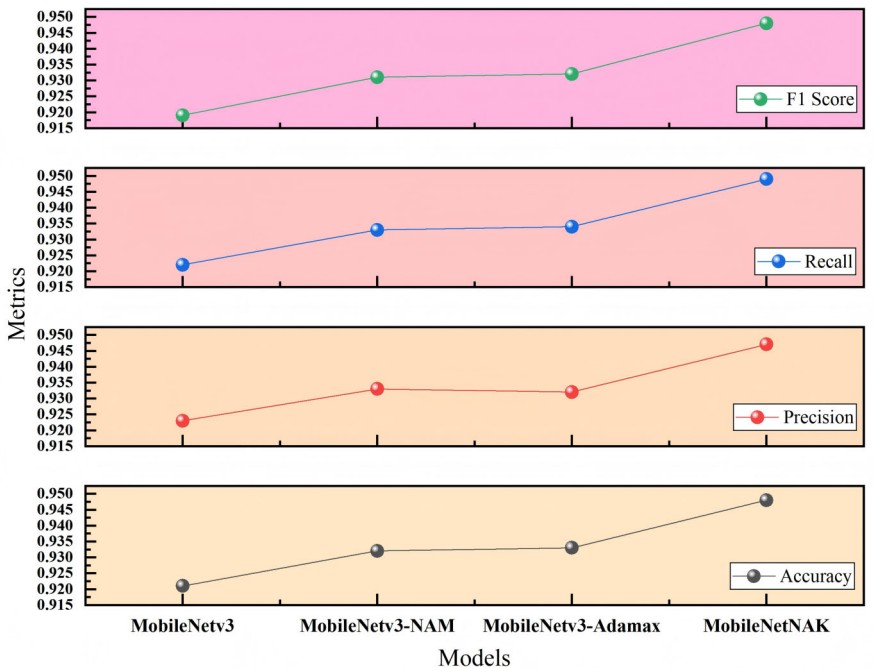

Figure 10 Ablation study results on Dataset 2.     

Table 3 Ablation results on Dataset 1 (Furniture age classification dataset).

| NAM | Adamax | KANs | Accuracy | Precision | Recall | F1-score | FPS |
|------|------|------|------|------|------|------|------|
| × | × | × | 0.837 | 0.836 | 0.834 | 0.835 | 213.54 |
| ✓ | × | × | 0.874 | 0.871 | 0.869 | 0.870 | 119.98 |
| ✓ | ✓ | × | 0.893 | 0.891 | 0.890 | 0.890 | 159.64 |
| ✓ | ✓ | ✓ | 0.904 | 0.902 | 0.900 | 0.901 | 147.80 |

Table 4 Ablation results on Dataset 2 (Bonn furniture style dataset).

| NAM | Adamax | KANs | Accuracy | Precision | Recall | F1-score | FPS |
|------|------|------|------|------|------|------|------|
| × | × | × | 0.921 | 0.923 | 0.922 | 0.919 | 213.54 |
| ✓ | × | × | 0.932 | 0.933 | 0.933 | 0.931 | 119.98 |
| ✓ | ✓ | × | 0.933 | 0.932 | 0.934 | 0.932 | 159.64 |
| ✓ | ✓ | ✓ | 0.948 | 0.947 | 0.949 | 0.948 | 147.80 |

Tables 3 and 4 present the ablation results conducted on two different datasets: the Furniture Age Classification Dataset (Dataset 1) and the Bonn Furniture Style Dataset (Dataset 2). The experiments focus on evaluating the performance impact of three key improvement modules proposed in this article: the NAM, the adaptive optimizer

(Adamax), and the KAN. Different combinations of these modules are used to assess their effect on model performance. A "×" indicates the module is not used, while a "✓" indicates it is applied. Performance metrics include accuracy, precision, recall, and F1-score.

As shown in Tables 3 and 4, for Dataset 1, the baseline model achieves significant improvement after integrating the NAM module, with accuracy increasing to 0.874. This indicates that the NAM module can effectively enhance the model's ability to capture key features such as wear and usage traces. With the further addition of the Adamax optimizer, accuracy increases to 0.893, verifying its advantages in training stability and convergence efficiency. The final MobileNetNAK model achieves an accuracy of 0.904 and an F1-score of 0.901. Compared with the baseline model, the overall performance improves significantly, demonstrating the synergistic optimization effect of the three modules.

For Dataset 2, a more challenging furniture style classification task, the accuracy of the baseline model rises to 0.932 after incorporating the NAM module. With the addition of the Adamax optimizer, accuracy slightly improves to 0.933. When all three improvements are applied, the accuracy reaches 0.948. Although the performance improvement is smaller than that in Dataset 1, it still shows good gains in the recognition of six furniture categories with 17 complex styles, further validating the generalization ability and stability of the proposed method across different task contexts.

The ablation results on both datasets fully demonstrate that: the NAM module significantly improves the model's ability to extract key features; the Adamax optimizer enhances training stability and convergence speed; and the KANs module strengthens the modeling of nonlinear features. The MobileNetNAK model, which integrates all three modules, performs excellently under various data distributions and task difficulties, verifying the generality and effectiveness of the proposed method.

In the experiments, to further evaluate the computational efficiency of the model during inference, we conducted five independent inference tests for each model configuration after training was completed. The average inference speed was calculated to obtain a stable FPS value. As shown in the results, the baseline MobileNetV3 achieved the highest inference speed, with an average of 213.54 fps, demonstrating the efficiency of its lightweight architecture. After integrating the NAM module, the inference speed decreased to 119.98 fps. With the addition of the Adamax optimizer, the FPS increased to 159.64. Finally, after incorporating the KANs module, the MobileNetNAK model achieved an average FPS of 147.80, still maintaining a high inference efficiency and effectively balancing accuracy with computational performance.

Although the proposed modules introduced a slight increase in computational overhead, they significantly improved classification performance while maintaining favorable real-time inference speed, indicating strong potential for practical deployment.

## Experimental results

Figure 11 illustrates the confusion matrix results on Dataset 1, comparing the classification accuracy and misclassification patterns across three categories: "new," "nearly new," and

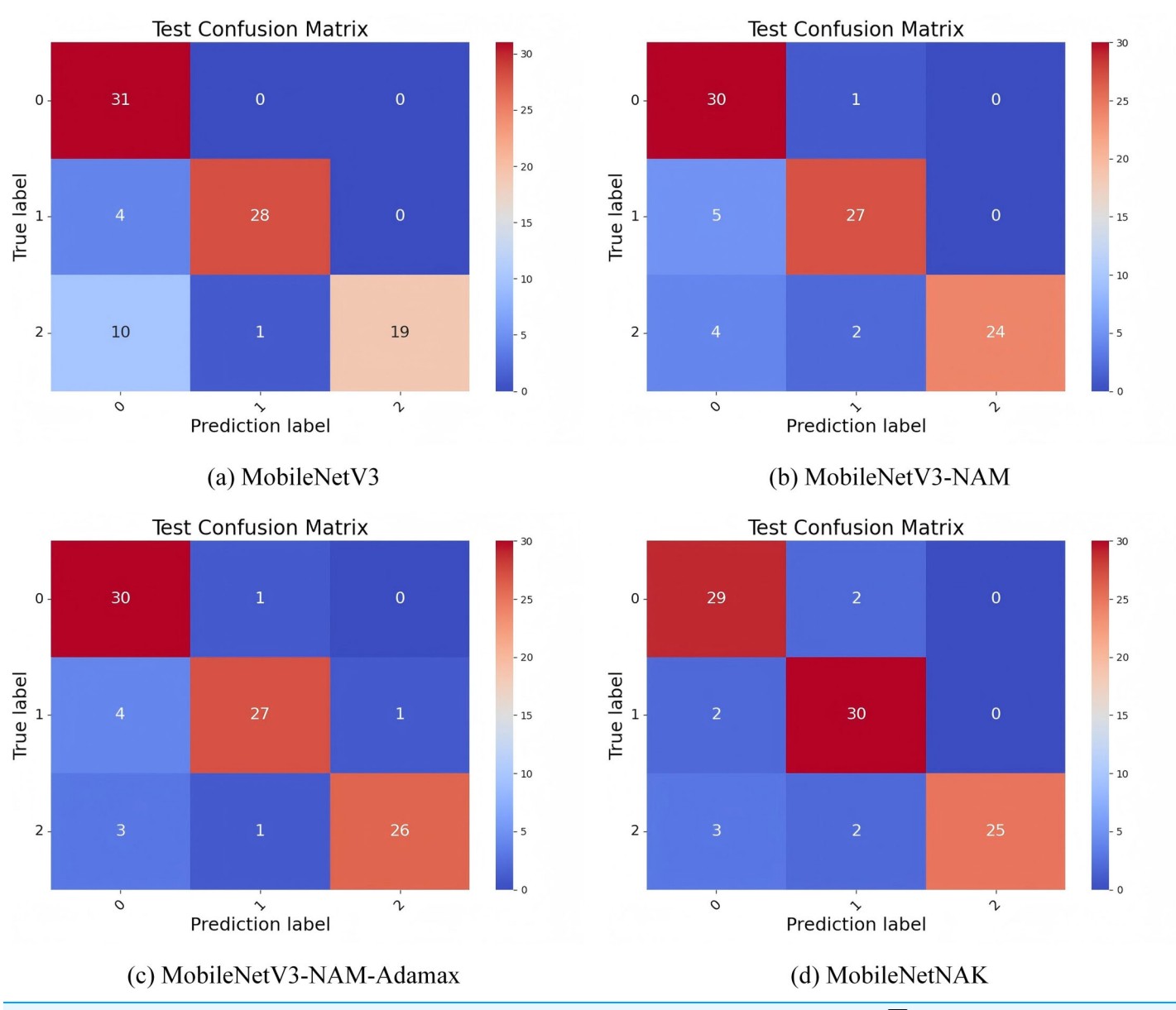

**Figure 11 Confusion matrix performance under different improvement strategies.**

"old." In the baseline MobileNetV3 model, the diagonal values corresponding to correct classifications are 31, 28, and 19, respectively. This indicates a noticeably lower accuracy in recognizing the "old" category, suggesting considerable confusion between similar classes.

With the introduction of the NAM module, the model's ability to classify "old" furniture improves significantly, increasing the number of correctly classified samples from 19 to 24. This improvement is attributed to the normalized attention mechanism embedded in NAM, which enhances the model's focus on critical regions—particularly those exhibiting usage traces and localized damage—thus improving the distinction between "nearly new" and "old" items.

Building on this, the incorporation of the Adamax optimizer further increases the correct predictions for the "old" class to 26. This confirms Adamax's effectiveness in stabilizing the training process and mitigating gradient oscillations, allowing the model to better capture fine-grained differences between categories and enhancing its generalization capability.

Finally, with the integration of the KANs module, the MobileNetNAK model achieves correct classification counts of 29, 30, and 25 for the "new," "nearly new," and "old" categories, respectively, demonstrating improved overall balance. By employing learnable univariate spline functions, KANs enhance the model's nonlinear representation capacity, particularly for subtle visual cues such as aging textures and wear patterns, which are crucial for fine-grained classification. Through the combined use of NAM, Adamax, and KANs, MobileNetNAK exhibits significantly improved class discrimination and overall robustness in furniture image classification tasks, especially in differentiating between visually similar categories.

Figure 12 illustrates the confusion matrix results on Dataset 2 under various model enhancement strategies, aiming to analyze the strengths and limitations of each model in multi-class furniture image recognition tasks.

In the baseline MobileNetV3 model, substantial misclassification is observed for the "chairs" category: only 81 samples are correctly classified, while 17 samples are erroneously predicted as "sofas" and 4 as "tables". This indicates a clear challenge in distinguishing between classes with similar structural or stylistic features. With the integration of the NAM, the model's recognition performance is significantly improved for categories such as "dressers", "beds", and "sofas". Specifically, the classification accuracy for "chairs" increases from 81 to 93, and for "dressers" from 104 to 105. NAM enhances the model's ability to focus on key structural regions within an image by combining channel and spatial attention mechanisms. Leveraging the sparsity of batch normalization scaling factors, the module improves feature selectivity and reduces misclassification among categories with similar textures.

Building upon this, the inclusion of the Adamax optimizer further enhances the model's performance, particularly for the "tables" category, where the number of correctly classified samples rises to 97. Moreover, non-diagonal entries for "chairs" and "dressers" become noticeably sparser. By employing the infinity norm for estimating the second moment of gradients, Adamax improves training stability and convergence speed, thereby enhancing generalization, especially under imbalanced class distributions.

Finally, the MobileNetNAK model, incorporating all three enhancements—NAM, Adamax, and KANs—achieves the most concentrated diagonal values across all categories. The model exhibits substantially improved discrimination for visually similar classes such as "chairs", "tables", and "sofas", with the respective correct classification counts increasing to 94, 95, and above 95. The integration of KANs introduces learnable univariate spline mappings to replace traditional linear weights, thus enhancing the model's nonlinear representation capacity. This is particularly effective in

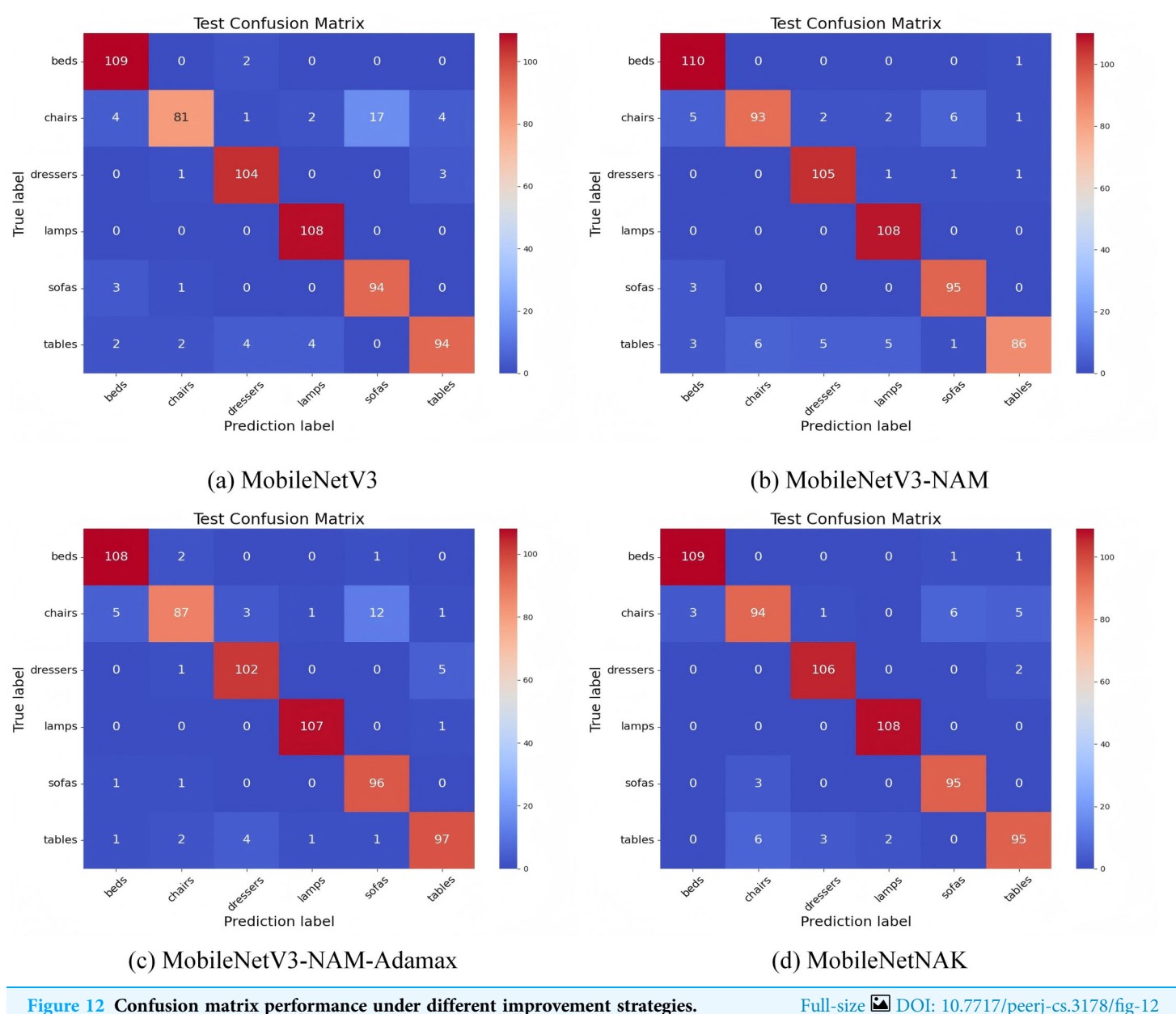

(a) MobileNetV3

(b) MobileNetV3-NAM

(c) MobileNetV3-NAM-Adamax

(d) MobileNetNAK

**Figure 12 Confusion matrix performance under different improvement strategies.**

capturing complex textures, contour variations, and fine-grained structural details in furniture images.

In summary, the three proposed modules—NAM, Adamax, and KANs—jointly contribute to a complementary synergy in visual attention, training optimization, and nonlinear feature modeling. Their integration not only significantly reduces cross-category misclassification in challenging cases but also improves the overall performance and stability of the model in fine-grained furniture image recognition tasks. These results comprehensively validate the effectiveness and generalizability of the proposed MobileNetNAK framework.

## CONCLUSIONS

This article deeply studies the task of furniture image classification, aiming to improve the classification performance by introducing innovative technical means. Based on the full investigation and analysis of the existing algorithms, this article proposes an improved algorithm MobileNetNAK based on MobileNetv3. The algorithm combines NAM module, Adamax optimizer and KANs, so that the model can capture the key features in the image more accurately, which not only accelerates the convergence speed of the model, but also improves the stability and classification performance of the training. It can further enrich the feature representation ability of the model, which can capture more subtle and complex feature information in the image, thereby further improving the classification effect. The joint application of these techniques significantly enhances the feature extraction ability and classification performance of the model. Through detailed experimental verification, the MobileNetNAK algorithm has achieved significant performance improvement in the furniture image classification task, which verifies the effectiveness of each technical means and the synergy between them. MobileNetNAK not only performs well in overall classification accuracy, but also has significant advantages in accurate identification of positive cases, comprehensive recall and comprehensive balance. Moreover, while maintaining high classification accuracy, MobileNetNAK achieves an inference speed of 147.80 FPS, demonstrating excellent computational efficiency and real-time responsiveness. The proposed method is not only suitable for academic evaluation scenarios but also exhibits strong deployment potential in practical applications within the furniture industry, particularly in scenarios such as intelligent quality inspection, automatic categorization, and online recommendation, where both real-time performance and accuracy are crucial. The research in this article not only provides a new solution for furniture image classification, but also provides a useful reference for image classification tasks in other related fields, showing its application potential and promotion value in complex image classification tasks.

### Funding

This research is supported by grant No.11901325 from the National Natural Science Foundation of China. Participated in the preliminary discussions of research design and provided software tools for data analysis.

### Grant Disclosures

The following grant information was disclosed by the authors:
National Natural Science Foundation of China: 11901325.

### Competing Interests

The authors declare that they have no competing interests.

## Author Contributions

- Danyang Zhang conceived and designed the experiments, analyzed the data, performed the computation work, prepared figures and/or tables, authored or reviewed drafts of the article, and approved the final draft.
- Yi Zhai performed the experiments, prepared figures and/or tables, and approved the final draft.
- Peiyuan Li conceived and designed the experiments, analyzed the data, performed the computation work, authored or reviewed drafts of the article, and approved the final draft.
- Fan Yang performed the experiments, prepared figures and/or tables, and approved the final draft.
- Runpeng Du conceived and designed the experiments, performed the computation work, authored or reviewed drafts of the article, and approved the final draft.

## Data Availability

The code is available in the Supplemental Files.

The dataset for furniture image classification is available at figshare: Li, Peiyuan (2025). archive. figshare. Dataset. https://doi.org/10.6084/m9.figshare.28831787.v2.

## Supplemental Information

Supplemental information for this article can be found online at http://dx.doi.org/10.7717/peerj-cs.3178#supplemental-information.

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
