# Peer review of "Research on furniture image classification based on MobileNetNAK"

_PeerJ Computer Science, doi:10.7717/peerj-cs.3178_

## Round 0.1 · original submission · Major Revisions

**Language Note:** The review process has identified that the English language must be improved. PeerJ can provide language editing services - please contact us at [email protected] for pricing (be sure to provide your manuscript number and title). Alternatively, you should make your own arrangements to improve the language quality and provide details in your response letter. – PeerJ Staff

Reviewer 1 ·

Basic reporting

1. Language and Expression:
The manuscript is generally well-written in English, but some sentences suffer from grammar and clarity issues. Professional English editing is recommended.

2. Dataset:
The dataset used is quite small and limited. Comparison with larger datasets such as "bonn furniture style" is advised. For a publication at this level, the dataset should be expanded.

3. Model Selection and Comparisons:
Experiments should be conducted not only with MobileNetV3 but also with other strong and modern models like Xception, ResNet, and EfficientNet. Improvements on MobileNetV3 may not generalize across all datasets.

4. Ablation Study:
The ablation study was only conducted internally within the proposed method. It should also be evaluated on other algorithms and different datasets.

5. Experimental Setup Details:
Information on the experimental environment (hardware, software, library versions, etc.) is missing and should be provided.

6. MobileNetV3 Usage:
Was MobileNetV3 tested in its vanilla form? Were layers fine-tuned or transfer learning applied? These details should be clarified.

7. Statistical Analysis:
Statistical significance tests (e.g., p-values) are missing from the results. Including such analyses would strengthen the validity of the findings.

8. Confusion Matrix Analysis:
A deeper analysis of the confusion matrices is needed to explain why the model performs poorly on certain classes.

9. Figure and Table Quality:
Figures and tables should be provided with higher resolution and better clarity.

Experimental design

The method used in the research should be given more clearly and compared with other CNN architectures.

Validity of the findings

-

Additional comments

1. Basic Reporting
• Language and Expression: Although the article is generally written in professional English, there are some grammatical errors and problems with clarity. For example, the sentence "This approach aims to boost the model's capability in identifying essential characteristics and enhancing classification precision" could be expressed more clearly.
• Structure: The article is structured in accordance with PeerJ standards, with logically separated sections. However, some subheadings could benefit from being more descriptive.
• Figures and Tables: Figures and tables are appropriately labeled and referenced in the text. However, some figures appear to be of poor quality (e.g., Figure 1 and Figure 2). It is recommended to provide higher resolution images.

2. Experimental Design
• Research Question: The research question is clearly defined, and it is stated that it aims to fill a gap in furniture image classification. However, more details could be provided on how this gap has been defined in the literature.

• Methods: The methods are well described and appear reproducible. However, it would be useful to provide more justification for why techniques such as NAM, Adamax, and KANs were chosen.

• Dataset: The dataset used (nearlyNewFurniture, newFurniture, oldFurniture) is described, but more information is missing about how the dataset was collected and labeled. It is also not stated whether the dataset is unbalanced.

3. Validity of the Findings

• Data Analysis: The results are supported by metrics such as Accuracy, Precision, Recall, and F1 Score, and verified with comparative experiments. However, statistical significance tests (e.g., p-values) are not presented. This makes the validity of the findings questionable.

• Ablation Study: The ablation study clearly showed the contribution of each component (NAM, Adamax, KANs). This is a strong aspect in terms of proving the effectiveness of the method.

• Confusion Matrix: Confusion matrices show in which classes the model performs worse. However, an analysis of why these errors occur is lacking.

4. General Comments
• Innovation: MobileNetNAK offers a significant innovation in MobileNetV3 by adding techniques such as NAM, Adamax, and KANs. However, a discussion of whether these techniques have been used in other studies is lacking.

• Applicability: Practical applications of the method in the furniture industry are discussed, but no assessment of how it would perform in real-world scenarios is provided.

• Literature Review: Relevant studies have been extensively reviewed, but it is thought that some important studies may have been missed. For example, more references could be made to similar furniture classification studies published in recent years.

5. Confidential Notes to the Editor
• The paper makes a significant contribution to the field of furniture image classification, and the experimental results are impressive. However, it can be accepted for publication if the above-mentioned deficiencies are addressed.

• Authors are advised to seek support from a professional editor, especially regarding language and expression.
Suggestions:
• Professional English editor support should be sought to correct language and grammar errors.

• More details should be added regarding the collection and labeling process of the dataset.

• Statistical significance tests (e.g., p-values) should be added to strengthen the validity of the findings.

The reasons for the errors in the confusion matrices should be analyzed and discussed.

The quality of the figures and tables should be improved, and higher resolution images should be provided.

Conclusion:
• The article has great potential as it presents a significant innovation, and the experimental results are impressive. However, it can be accepted for publication if the above suggestions are taken into consideration. I recommend a "Major Revision".

The dataset is quite small, compared with Bonn Furniture Style. The dataset is very small for a publication of this level.

Algorithms such as Xception, ResNet, and EfficientNet should be tried. Adding them via mobilenetv3 may not give the same result for all data.

Ablation has been done within itself. It should be tried on other algorithms and data sets.

There is no information about where and in what environment the experiments were conducted.
Was Mobile Netv3 tried in its bare form? Were the layers opened for training? Was transfer learning used?

Cite this review as

Reviewer 2 ·

Basic reporting

1. It would be better for the authors to add references to some of the latest methods. For example, "CVANet: Cascaded Visual Attention Network" and "GACNet: Generate Adversarial-driven Cross-aware Network", which are closely related to your manuscript, especially in terms of advanced object recognition or classification network architectures.

2. It is recommended that the authors clearly summarize the main contributions of the paper. While the article provides a systematic review, it lacks a focused synthesis of the core contributions. The authors are encouraged to list three major contributions in a bullet-point format to enhance clarity.

Experimental design

3. The sizes of variables in the article formulas are inconsistent, as seen in the variable "q" in formulas (5) and (6). Additionally, variable names for specific terms should be in regular font, not italics. The author is requested to check and correct this carefully.

4. The number of comparative experiments in the article is relatively small, and the comparative methods used are relatively outdated. It is suggested to supplement the latest research methods from the past three years to highlight the innovation and progressiveness of the techniques you have proposed.

5. Although the experimental results show that the method proposed by the author performs well in various indicators, there is a lack of in-depth analysis of the specific reasons for its performance improvement. Suggest that the author add relevant discussions.

Validity of the findings

6. The author mentions in the abstract that the introduced modules have improved computational efficiency; however, there is a lack of specific experimental verification of computational efficiency in the experimental section. Suggest that the author supplement relevant time performance experiments to enhance the completeness and persuasiveness of the paper.

7. The method proposed by the author has indeed demonstrated some effectiveness; however, in terms of innovation, compared to the latest research results in the field of deep learning, this study falls slightly short.

Additional comments

This article proposes a furniture image classification algorithm called MobileNetNAK, which is based on the MobileNetV3 network and combines the NAM module, Adamax optimizer, and Kolmogorov-Arnold Networks (KANs) technology. Firstly, the NAM module significantly enhances the model's ability to extract key image features by introducing normalization operations and attention mechanisms. Secondly, the Adamax optimizer improves the convergence speed and stability during the training process by adaptively adjusting the learning rate. Finally, KANs have improved the computational efficiency and feature extraction capability of the model by replacing complex convolution operations. The experimental results demonstrate that this method outperforms previous methods in terms of accuracy; however, it also presents some issues.

Cite this review as

---

## Round 0.2 · accepted · Accept

Congratulations on having addressed all concerns.

Reviewer 1 ·

Basic reporting

The research has been addressed, and the work has become quite successful. It can be published.

I congratulate the authors on their efforts.

Experimental design

-

Validity of the findings

-

Cite this review as

Reviewer 2 ·

Basic reporting

Clear

Experimental design

Clear

Validity of the findings

Clear

Cite this review as